evolution, genetics

*Lotus japonicus*, experimental evolution, microbiome engineering, rhizobia, mutualism, symbiosis

**Author for correspondence:**
Joel L. Sachs
e-mail: joels@ucr.edu

# Experimental evolution can enhance benefits of rhizobia to novel legume hosts

Kenjiro W. Quides[1], Alexandra J. Weisberg[4], Jerry Trinh[1], Fathi Salaheldine[1], Paola Cardenas[1], Hsu-Han Lee[1], Ruchi Jariwala[1], Jeff H. Chang[4] and Joel L. Sachs[1,2,3]

[1]Department of Evolution, Ecology, and Organismal Biology, [2]Department of Microbiology and Plant Pathology, and [3]Institute for Integrative Genome Biology, University of California, Riverside, CA, USA
[4]Department of Botany and Plant Pathology, Oregon State University, Corvallis, OR, USA

 KWQ, 0000-0003-2015-5264; AJW, 0000-0002-0045-1368; JHC, 0000-0002-1833-0695; JLS, 0000-0002-0221-9247

Legumes preferentially associate with and reward beneficial rhizobia in root nodules, but the processes by which rhizobia evolve to provide benefits to novel hosts remain poorly understood. Using cycles of *in planta* and *in vitro* evolution, we experimentally simulated lifestyles where rhizobia repeatedly interact with novel plant genotypes with which they initially provide negligible benefits. Using a full-factorial replicated design, we independently evolved two rhizobia strains in associations with each of two *Lotus japonicus* genotypes that vary in regulation of nodule formation. We evaluated phenotypic evolution of rhizobia by quantifying fitness, growth effects and histological features on hosts, and molecular evolution via genome resequencing. Rhizobia evolved enhanced host benefits and caused changes in nodule development in one of the four host–symbiont combinations, that appeared to be driven by reduced costs during symbiosis, rather than increased nitrogen fixation. Descendant populations included genetic changes that could alter rhizobial infection or proliferation in host tissues, but lack of evidence for fixation of these mutations weakens the results. Evolution of enhanced rhizobial benefits occurred only in a subset of experiments, suggesting a role for host–symbiont genotype interactions in mediating the evolution of enhanced benefits from symbionts.

## 1. Introduction

Plants interact with diverse soil microbes that can enhance their health and fitness [1,2]. However, both host and microbe genotypes and the environment can influence microbial symbioses, and fitness outcomes can range from mutualistic to parasitic for the host plant [2–6]. Plants have evolved a suite of 'host control' traits that bias *in planta* resources towards cooperative microbial genotypes and defend against or sanction harmful ones [3,7–9]. Moreover, plants often exhibit segregating variation for these traits, suggesting that host control can be shaped by natural or artificial selection [2,10–12]. But we do not understand how plant hosts—and specific host control traits—impose selection on soil microbes, which is vital to resolving how bacterial mutualists evolve, and informing on applications to leverage their services [13].

Plants can impose selection for beneficial strains among microbes that are available in the soil [14–17], but it is unclear if this leads to lasting change in the soil microbiome [1,13]. Microbes have an evolutionary advantage over hosts in terms of population size and generation time, leaving hosts vulnerable to exploitation [17]. Microbes can also experience free-living phases in the soil with a different set of selection pressures [18–20]. Under selection by the host and in the soil, microbes can evolve enhanced services for the host, phenotypes that exploit the host or traits that increase fitness in the soil between phases of host interaction [4].

The legume–rhizobia association has a bipartite life cycle like that of many other horizontally acquired microbial symbioses, with phases of host infection

alternating with free-living periods in the environment [6,21]. An exchange of signals between host and symbiont triggers the formation of nodules and the intracellular colonization of root tissue, wherein compatible rhizobia initiate nitrogen fixation [22]. After nodules develop, hosts can detect the net benefit that rhizobia provide and can sanction rhizobia that do not fix sufficient nitrogen through the selective senescence of the nodule cells they reside within [23]. The entire nodule eventually breaks down as host resources are redirected to seed production and a subset of rhizobia are released back into the soil [24]. Host control, occurring at multiple stages of the symbiosis, is thought to be critical in imposing selection for beneficial rhizobia *in planta*.

Here, we investigated the capacity for selection imposed by legumes—over multiple growing cycles—to lead to rhizobia with enhanced beneficial effects on the model legume *Lotus japonicus*. Using a full-factorial replicated design, we experimentally evolved two rhizobia strains in independent associations with each of two *Lotus japonicus* genotypes that vary in regulation of nodule formation. Hosts included the MG-20 ecotype and a near isogenic hypernodulating mutant, *har1*, which can form as many as six times the nodules of MG-20 [25]. In separate experiments, infections were initiated with either *Rhizobium etli* CE3 or *Ensifer fredii* NGR234, strains that provide marginal benefit to *L. japonicus* [26,27]. CE3 is a beneficial symbiont of *Phaseolus vulgaris* [28], but on *L. japonicus* nodules undergo premature senescence, hindering nitrogen fixation as early as three weeks post inoculation (wpi) [26]. NGR234 (hereafter NGR) nodulates diverse legumes, including *L. japonicus* [29]. When NGR infects MG-20, nodule development is delayed. NGR slowly ramps up nitrogen fixation over the course of 8–12 weeks and does not reach a maximum until 20 wpi [27]. After experimental evolution, derived rhizobia populations were compared with their ancestral genotypes, using inoculation experiments, *in vitro* fitness analysis, nodule histology and via whole genome resequencing. The goals were to (i) investigate the symbiont phenotypes that evolve after recurrent cycles of host infection, including effects on the host and fitness during free-living phases, (ii) identify genomic changes that occurred during symbiont evolution, and (iii) compare evolutionary outcomes in the context of different host and rhizobia genotypes.

## 2. Material and methods

### (a) Experimental evolution protocol

Seeds were surface sterilized, nick scarified and placed into sterilized CYG germination pouches (Mega International) filled with 20 ml of sterile nitrogen-free Jensens fertilizer [30]. Plants were maintained in a growth chamber with a light : dark cycle of 14 : 10 h at approximately 600 Lux, 18–27.5°C, and relative humidity of 40–65%. When seedlings had at least two true leaves (approx. 2.5 weeks after planting), 50 μl of $5.0 \times 10^7$ rhizobia cells were dripped directly on roots. For the initial round of infection (passage 0) plants were infected with liquid cultures of CE3 or NGR (ancestral clones). Subsequent rounds of inoculation were initiated from rhizobia extracted from the nodules of the previous passage (descendant populations). Plants were fertilized weekly with 10 ml of nitrogen-free Jensens fertilizer per pouch. Each of the four host and symbiont combinations were passaged in two duplicates for a total of eight experiments.

At 4 wpi of passaging, nodules were counted, dissected and photographed, and shoots were dried to weigh biomass. To extract rhizobia, dissected nodules were pooled by experiment and surface sterilized in bleach, rinsed in sterile water, macerated and resuspended in 5 ml of a modified arabinose gluconate media; MAG [31] (figure 1). From each nodule extract, 3.2 ml was used to inoculate a flask of MAG to grow cells for the next passage. MAG is a relatively low nutrient medium, with 1 gm l$^{-1}$ of yeast, or about 20% compared to rich media [31]. This *in vitro* growth phase represents a free-living state experienced by rhizobia between host infections and allowed us to inoculate the next round of hosts with a consistent number of cells every passage. From 5 ml of nodule extracts, 200 μl was mixed with 200 μl of MAG : glycerol (1 : 1) to archive cells and 100 μl was serially diluted ($10^{-6}$) and plated on MAG to quantify rhizobia population size within nodules.

CE3 was evolved for 15 passages in all cases. The NGR passage lines became contaminated at passage 3 (*har1* hosts) and passage 13 (MG-20 hosts), and a new round of passaging was initiated from stocks archived prior to contamination. As a result of contamination, NGR was only passaged for 10 cycles on *har1*. Because of the reduced number of passages, both replicates of the NGR:*har1* combination were exposed to 80 plants in passages 6–10 (compared to 40 in previous passages), resulting in a total of approximately 450 plants compared to approximately 260 in the other passage lines.

Founding population sizes of nodules (i.e. infection bottleneck) were estimated using a model parametrized with empirical data; inoculation with $5 \times 10^7$ cells was predicted to generate approximately 6% of coinfected nodules [32], resulting in an estimated bottleneck of 1.06 per nodule multiplied by the total number of nodules formed on each cohort of inoculated hosts (i.e. approx. 40–80 plants). Final population sizes of rhizobia (when nodules were dissected for passaging) were quantified for each passage through serial dilutions and spread plating of rhizobia cultured from nodules at 4 wpi. Initial and final rhizobial population sizes were averaged over passages with standard error. Initial and final *in planta* population sizes were used to calculate the number of generations *in planta*. The numbers of *in vitro* generations were calculated using 64% of the total *in planta* population size of rhizobia (i.e. 3.2 ml divided by 5 ml) as the starting population size, and the total number of cells within the flask after the period of *in vitro* growth, estimated via optical density.

### (b) Phenotypic analyses

#### (i) *In vitro* growth rate and cell density estimation
Individual colonies from ancestral and derived populations were inoculated into liquid MAG and grown to log-phase (12–36 h, 29°C, 180 r.p.m.). Doubling times were calculated between sequential time points using 12–28 replicate flasks per estimate.

#### (ii) Symbiont effectiveness and fitness *in planta*
To measure evolutionary changes in symbiont effectiveness on hosts, seedlings were planted in sterilized Conetainers (SC10; Steuwe and Sons) filled with autoclaved inert calcined clay (Pro League; Turface Athletics), and grown in a controlled facility with daily mist-watering until true leaves formed and thereafter were fertilized weekly with 5 ml of nitrogen-free Jensens solution. Seedlings were transferred to the greenhouse, hardened behind 50% shade cloth for 4 days, grouped by size within each genotype (using leaf counts), and seedlings from each group were randomly assigned to inoculation treatments. Each host genotype (MG-20, *har1*) received inoculation treatments, including the ancestral or derived rhizobial populations from each rhizobia strain (CE3, NGR; $5 \times 10^8$ cells in 5 ml, dripped directly onto soil) and experimental replicate (a, b), or water as a control. The 14 host and inoculum treatment combinations were organized into blocks and were randomly distributed in

## experimental evolution

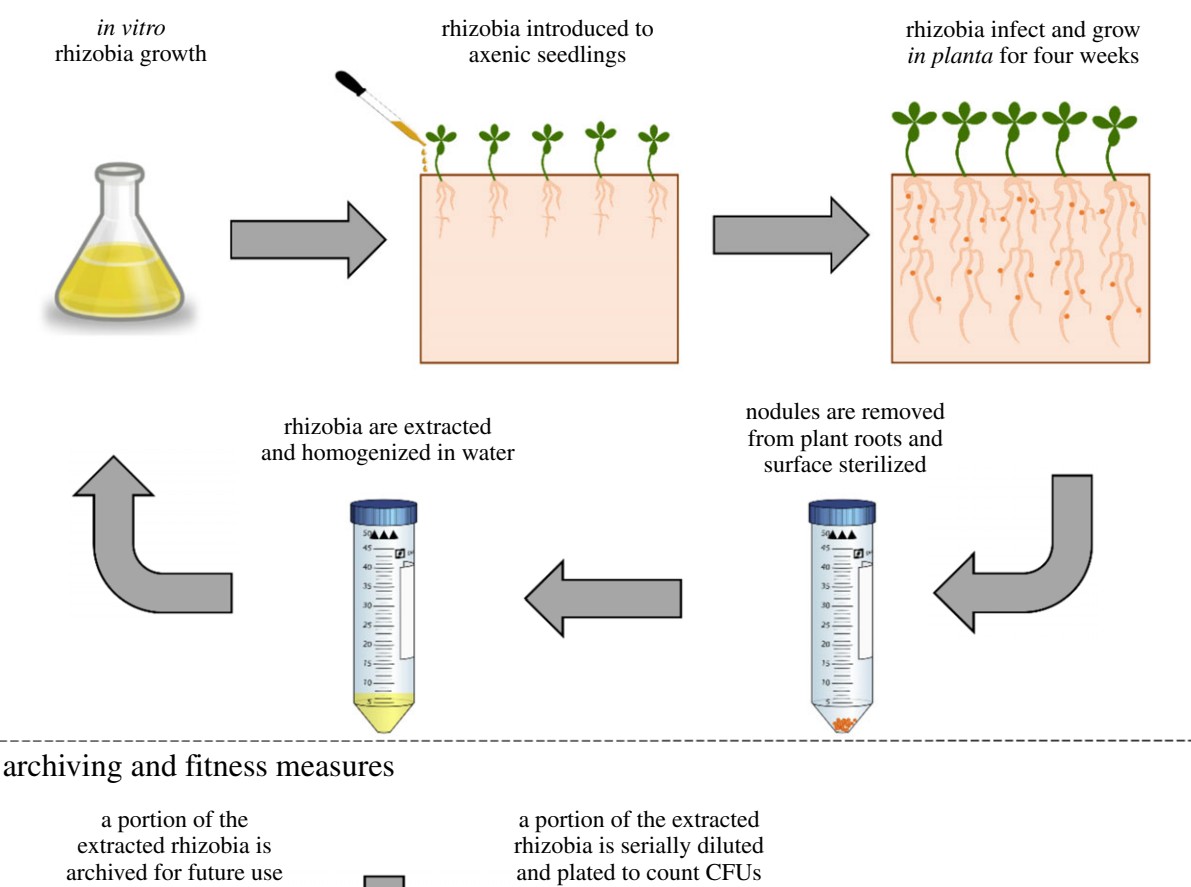

**Figure 1.** The experimental evolution protocol allows microbes—but not plants—to evolve. Rhizobia are grown *in vitro,* and $5.0 \times 10^7$ rhizobia cells are inoculated directly on axenic plant roots. Plants are grown for 4 wpi, after which nodules are removed, and rhizobia are extracted to start a new round of *in vitro* growth. A portion of the extracted rhizobia are archived for future experiments. Another portion of the extracted rhizobia are serially diluted to quantify *in planta* population sizes to estimate the number of *in planta* and *in vitro* generations. (Online version in colour.)

the greenhouse with one treatment combination replicate in each of the 20 blocks (280 plants total).

Ten blocks of plants were harvested at 4 wpi, matching the timepoint when selection was imposed (i.e. when rhizobia were extracted from plants for passaging). The remaining 10 blocks were harvested at 6 wpi to examine changes in timing of growth effects on hosts. At both harvest times, measurements were taken for shoot biomass, number of nodules formed, total nodule biomass and mean biomass of individual nodules. Tissues were dried at 60°C prior to weighing.

Population sizes of rhizobia in nodules, a proxy of fitness *in planta,* were estimated at both harvest times. From each treatment replicate, four randomly picked nodules per plant were used (144 total nodules per harvest). Surface sterilized nodules were macerated and serially diluted in sterile $H_2O$, spread plated on MAG ($10^{-3}$, $10^{-5}$ dilutions) and incubated at 29°C. For CE3, cultures were incubated overnight, and for NGR, cultures were incubated for 3 days.

### (iii) Nodule morphology
Plants were randomly selected for light microscopy analysis at the 6 wpi harvest, a key timepoint to analyse *L. japonicus* nodule

senescence [23]. Nodules were fixed in a paraformaldehyde-glutaraldehyde solution, before being infiltrated in JB-4 Plus methacrylate, following published protocols [23]. Using a glass knife and an H/I Bright 5030 Microtome (Hacker Instruments Inc.), nodule sections of 4 µm thickness were prepared parallel to the long axis of the parent root. Sections were mounted on glass slides and stained with 0.1% w/v aqueous toluidine Blue O, which stains acidic tissues, allowing for identification of infected plant cells.

An average of 16 sections per nodule were analysed (range 4–30) from each of three to four nodules per host plant. For each section, we calculated the percentage of infected plant cells that were ruptured with blotchy appearance and low densities of rhizobia, indicating nodule cell senescence [23]. To control for variation in cell structure throughout the nodule, the mean percentage of senescent plant cells in a section was calculated for all sections from one nodule [24].

### (iv) Nitrogen fixation by rhizobia
Using isotopic analysis, %N and $\delta^{15}N$ were quantified for all host and treatment combinations at both harvest times. When plants incorporate fixed nitrogen, leaf tissues exhibit a decrease in

$\delta^{15}$N relative to uninfected plants because of isotopic fractionation by rhizobia [33]. Dried leaves were removed from stems and powdered using a 5 mm bead beater for 10 s at 4 ms$^{-1}$. Samples were analysed at the UC Santa Cruz Stable Isotope Laboratory. We pooled leaf tissue from up to four plants in a treatment because single plants often did not provide sufficient tissue for analysis. Due to pooling, each treatment had two to five replicates.

### (v) Data analysis

Trait measures were transformed as necessary to improve normality. Trait comparisons, between ancestral and derived symbionts, were performed using Welch's two sample $t$-tests, allowing for unequal variance in the measures of %N and $\delta^{15}$N.

## (c) Genome sequence analysis

The CE3 ancestral clone and archived nodule slurries from the CE3 experiments were plated on MAG with 25 µg ml$^{-1}$ cyclohexamide. Twenty colonies each were sampled from the passage 15 populations of CE3:har1_a and CE3:har1_b (i.e. replicate experiments of CE3:har1) and one CE3 colony (ancestor). Individual colonies were picked and washed in 500 µL of sterile H$_2$O via vortexing and centrifugation (14 000$g$, 3 min). From passages 5 and 10, we created pooled samples of cells from CE3:har1_a and CE3:har1_b. From passage 15, we created pooled samples of cells from CE3:har1_a, CE3:har1_b, CE3:MG-20_a, and CE3:MG-20_b. For these eight populations of pooled samples, dense colonies on plates were scraped and washed in 10 ml of sterile H$_2$O. A total of 25 µl of washed population cell culture was used for DNA extraction. Genomic DNA was extracted using a DNeasy blood & tissue kit (Qiagen).

Genomic DNA was processed as Nextera XT libraries and sequenced (2× 150 bp) on one lane of an Illumina HiSeq 3000 at the Center for Genome Research and Biocomputing (CGRB), Oregon State University. Sequencing reads were processed and assembled and contigs were annotated as previously described [34].

Bowtie2 v. 2.2.3 with the option '-local' was used to map reads to the *R. etli* CFN 42 reference sequence (NCBI accession: GCF_000092045.1) [35]. Samtools v. 0.1.18 was used to convert alignments to bam format and Picard tools v. 2.0.1 was used to add sample read groups to alignments [36]. Mapped read coverage for each sample was calculated using bedtools v. 2.25.0 'genomecov', and regions with no coverage were identified as putative deletions in each sample [37]. GATK v. 3.7 HaplotypeCaller and the options '-ERC GVCF -ploidy 1' were used to call variants for each sample, and the data were then combined using GenotypeGVCFs [38]. SnpEff v. 4.3t and the CFN 42 reference genome sequence were used to annotate variants for predicted functional effects [39].

## 3. Results

### (a) Phenotypic evolution of rhizobia

After 15 generations of experimental evolution, we found evidence for enhanced host benefits in the CE3 symbionts. The combination of CE3 with *har1* hosts produced the most striking results, with both replicate CE3:*har1* populations showing enhanced growth benefits to hosts, indicated by a significant increase in shoot biomass at 4 wpi in plants inoculated with derived populations relative to the ancestral CE3 (table 1 and figure 2$a$). CE3 did not show evidence for the evolution of increased nitrogen fixation on hosts, as inoculation with derived CE3 populations did not cause hosts to differ in their percent nitrogen in leaf tissue, or isotopic

signature, relative to hosts inoculated with the CE3 ancestor (table 1). These data suggest that CE3 evolved to enhance host growth by decreasing costs to plants (i.e. metabolic requirements within nodules) as opposed to a change in the gross benefit to hosts (i.e. nitrogen fixation).

We examined timing of CE3 effects on the *har1* hosts including nodule senescence and growth benefits. Histological features of nodule morphology and senescence indicated that, contrary to predictions, CE3 evolved to induce more severe senescence on hosts. On the *har1* hosts at 6 wpi, a higher proportion of cells showed signs of senescence in nodules infected with the derived CE3 populations compared to the ancestral CE3, a pattern that was significant in the 'a' replicate population ($t = 4.228$, d.f. = 5.567; $p = 0.006$), and was similar for the 'b' replicate, but not significant ($t = 1.44$, d.f. = 2.99 $p = 0.245$; figure 3). Consistent with previous work [26], induction of nodule senescence halted growth benefits from symbiosis. A comparison of shoot growth rate per day between the 4 and 6wpi timepoints found that growth rate was decreased for *har1* hosts inoculated with the derived CE3 population compared to hosts inoculated with the ancestral CE3 (figure 2$b$), a pattern that was significant in the 'a' replicate population (ANCOVA: $F_{1,36} = 5.77$, $p = 0.022$), and was marginal in the 'b' population (figure 2$b$; ANCOVA: $F_{1,36} = 3.71$, $p = 0.06$). Moreover, there were no significant differences in growth effects between the CE3 ancestor and descendants at 6 wpi (table 1). These data suggest that selection favored rhizobia that enhanced host growth for the timepoint when selection was imposed (4 wpi), but not thereafter (6 wpi). Unlike on the *har1* host, we uncovered only marginal evidence for the evolution of enhanced host benefits for the CE3 populations that evolved on MG20 hosts (i.e. only detected in one replicate at the 6 wpi harvest; table 1).

We uncovered no support that NGR evolved to enhance shoot growth in NGR. Instead we uncovered differences in evolved populations that were inconsistent among experiments (table 1). For the NGR populations that evolved on *har1* hosts, we found evidence of increased leaf nitrogen coupled with decreased nodule mass in the 'a' replicate population (4 wpi), but increased nodule mass in the 'b' replicate population (6 wpi). For the NGR populations that evolved on MG20 hosts, we found evidence of decreased rhizobia population sizes in nodules in both replicate populations in addition to a decreased nitrogen fixation in the 'a' replicate population (4 wpi). These results suggest stochastic changes that are not consistent with selection for enhanced symbiotic benefits. Although NGR was only passaged for ten generations on *har1*, thus growing for fewer generations *in vitro*, these populations grew for a similar or greater number of generations *in planta* than the CE3 populations (table 2; electronic supplementary material, table S3).

### (b) Molecular evolution of CE3

Several protein-coding changes were uncovered among the 20 isolate samples that were sequenced in each of the evolved CE3:*har1*_a and CE3:*har1*_b replicate populations (table 3). Eight out of the 20 CE3:*har1*_a isolates had a missense mutation in a gene coding for sorbosone dehydrogenase, potentially enhancing rhizobial persistence in intercellular space [40]. Three of the 20 isolates had a missense mutation in the gene for alpha-ʟ-fucosidase, associated with nod factor production [41]. Three of the 20 CE3:*har1*_b isolates had a

**Table 1.** Statistics for comparisons between ancestral and derived states. Ancestral and derived phenotypes compared with Welch's two sample $t$-test. $t$-value compares derived relative to ancestral (ex., $t < 0$ indicates derived < ancestral). Asterisks (*) indicate $p \leq 0.05$ and ** indicates $p \leq 0.01$.

| | CE3: har1_a | CE3: har1_b | CE3: MG-20_a | CE3: MG-20_b | NGR: har1_a | NGR: har1_b | NGR: MG-20_a | NGR: MG-20_b |
|---|---|---|---|---|---|---|---|---|
| **phenotype (4 wpi)** | | | | | | | | |
| shoot biomass | $t = 3.15$** <br> d.f. = 14.1 | $t = 2.81$* <br> d.f. = 11.5 | $t = 0.429$ <br> d.f. = 16 | $t = 0$ <br> d.f. = 14.3 | $t = 0$ <br> d.f. = 16.7 | $t = -1$ <br> d.f. = 17.5 | $t = -1.32$ <br> d.f. = 16.6 | $t = -0.845$ <br> d.f. = 16.7 |
| percentage nitrogen in leaf tissue | $t = -0.277$ <br> d.f. = 1.9 | $t = -3$ <br> d.f. = 1 | $t = -0.632$ <br> d.f. = 2.44 | $t = 0.655$ <br> d.f. = 3.92 | $t = 4.58$ * <br> d.f. = 2.88 | $t = 2.4$ <br> d.f. = 1.17 | $t = 1$ <br> d.f. = 2 | $t = -1$ <br> d.f. = 2 |
| $\delta^{15}N$ | $t = 0.975$ <br> d.f. = 1.31 | $t = 1.45$ <br> d.f. = 3 | $t = -0.204$ <br> d.f. = 3.87 | $t = 0.35$ <br> d.f. = 3.4 | $t = -0.16$ <br> d.f. = 2.91 | $t = -4.11$ <br> d.f. = 2.07 | $t = -3.55$ * <br> d.f. = 3.44 | $t = 3.07$ <br> d.f. = 2.56 |
| total nodule mass | $t = 0.67$ <br> d.f. = 13.2 | $t = 0.114$ <br> d.f. = 13.9 | $t = -0.679$ <br> d.f. = 7.2 | $t = -0.216$ <br> d.f. = 11.1 | $t = -2.36$ * <br> d.f. = 15.4 | $t = -0.946$ <br> d.f. = 13.8 | $t = -0.362$ <br> d.f. = 13.6 | $t = -0.405$ <br> d.f. = 12.9 |
| log(estimated nodule population size) | $t = 0.163$ <br> d.f. = 16.7 | $t = 0.481$ <br> d.f. = 16.6 | $t = 0.226$ <br> d.f. = 20 | $t = -0.382$ <br> d.f. = 17.9 | $t = 0.797$ <br> d.f. = 17.2 | $t = 0.639$ <br> d.f. = 14.1 | $t = -2.59$ * <br> d.f. = 13.2 | $t = -2.53$ * <br> d.f. = 10.9 |
| percentage of senescing nodule cells | n.a. | n.a. | n.a. | n.a. | n.a. | n.a. | n.a. | n.a. |
| *in vitro* growth rate (doubling time) | $t = -0.984$ <br> d.f. = 65.2 | $t = -1.07$ <br> d.f. = 59.2 | $t = -1.19$ <br> d.f. = 60.7 | $t = -0.046$ <br> d.f. = 75.4 | $t = 0.305$ <br> d.f. = 177 | $t = 0.176$ <br> d.f. = 174 | $t = 0.099$ <br> d.f. = 146 | $t = 0.539$ <br> d.f. = 128 |
| **phenotype (6 wpi)** | | | | | | | | |
| shoot biomass | $t = -1.77$ <br> d.f. 13.4 | $t = -1.03$ <br> d.f. = 12.3 | $t = 2.33$* <br> d.f. = 17.4 | $t = 0.384$ <br> d.f. = 18 | $t = 0.177$ <br> d.f. = 17.2 | $t = 0.186$ <br> d.f. = 17.9 | $t = 0.12$ <br> d.f. = 14 | $t = -0.538$ <br> d.f. = 17.3 |
| percentage nitrogen in leaf tissue | $t = -0.2$ <br> d.f. = 1 | $t = 1$ <br> d.f. = 2 | $t = 1.1$ <br> d.f. = 4.41 | $t = 1.42$ <br> d.f. = 4.54 | $t = 0.459$ <br> d.f. = 3.74 | $t = 0.894$ <br> d.f. = 3.67 | $t = 0.454$ <br> d.f. = 2.68 | $t = -1.26$ <br> d.f. = 5.6 |
| $\delta^{15}N$ | $t = 0.439$ <br> d.f. − 1.25 | $t = -0.222$ <br> d.f. = 2.79 | $t = 0.492$ <br> d.f. = 4.03 | $t = -0.191$ <br> d.f. = 5.68 | $t = -0.792$ <br> d.f. = 2.84 | $t = -2.56$ <br> d.f. = 3.57 | $t = 1.86$ <br> d.f. = 3.56 | $t = -0.257$ <br> d.f. = 4.72 |
| total nodule mass | $t = -1.24$ <br> d.f. = 10.9 | $t = -0.448$ <br> d.f. = 16 | $t = 0.669$ <br> d.f. = 16 | $t = 0.574$ <br> d.f. = 15.8 | $t = 2.13$ <br> d.f. = 14.1 | $t = 2.5$ * <br> d.f. = 15.1 | $t = 0.594$ <br> d.f. = 15.5 | $t = 0.208$ <br> d.f. = 16 |
| log(estimated nodule population size) | $t = 3.02$** <br> d.f. = 16 | $t = 1.48$ <br> d.f. = 21 | $t = 0.238$ <br> d.f. = 18.9 | $t = 0.161$ <br> d.f. = 21.6 | $t = -0.207$ <br> d.f. = 18 | $t = -1.83$ <br> d.f. = 18.2 | $t = 0.734$ <br> d.f. = 21 | $t = -0.453$ <br> d.f. = 20.9 |
| percentage of nodule cells undergoing senescence | $t = 4.23$** <br> d.f. = 5.57 | $t = 1.44$ <br> d.f. = 2.99 | $t = 0.128$ <br> d.f. = 5.71 | $t = 0.804$ <br> d.f. = 4.06 | n.a. | n.a. | n.a. | n.a. |
| *in vitro* growth rate (doubling time) | see above | see above | see above | see above | see above | see above | see above | see above |

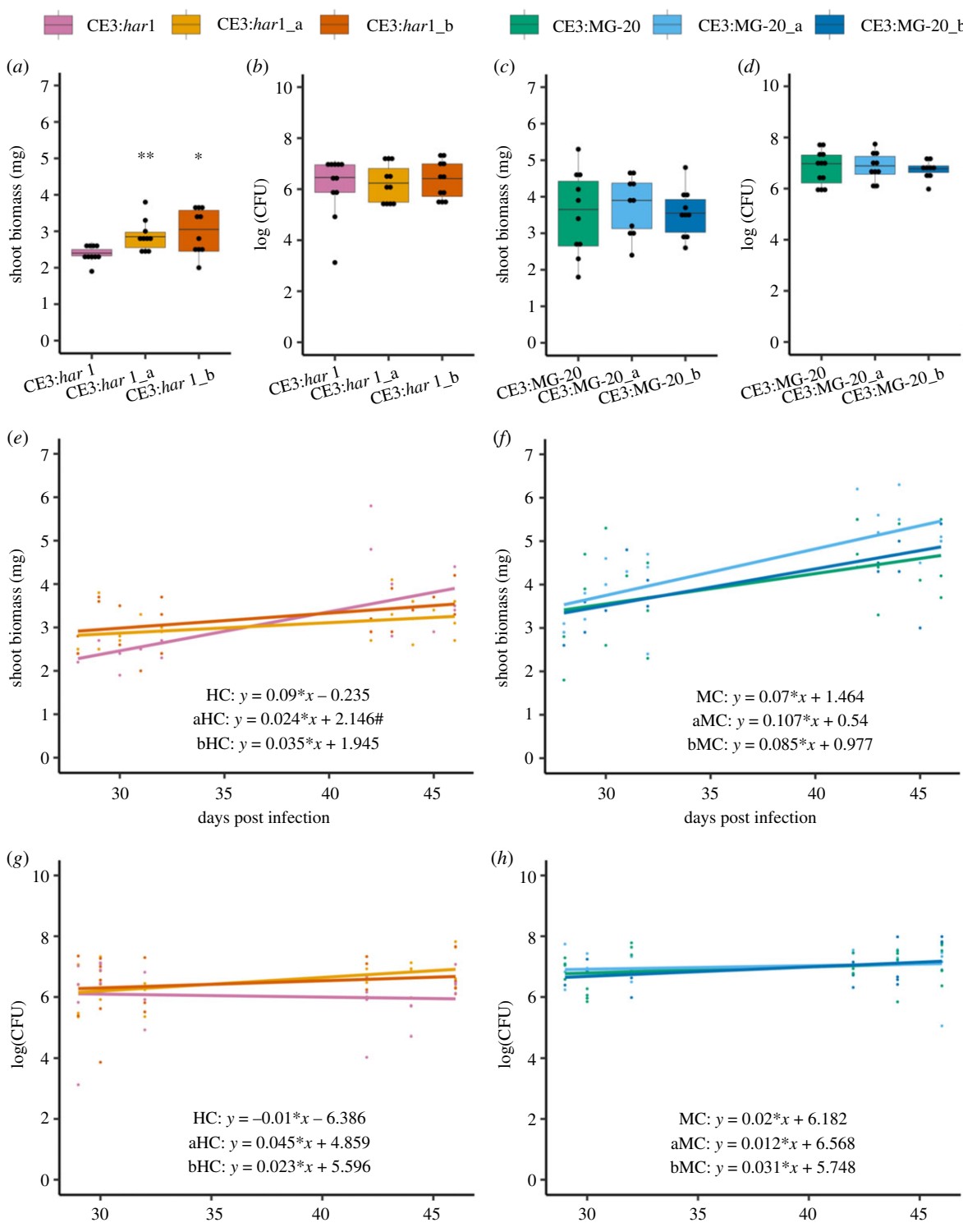

**Figure 2.** Phenotypic evolution of CE3 symbionts. Symbiotic effectiveness was measured as shoot biomass (*a,c,e,f*) and symbiont fitness was estimated using the mean population size of rhizobia within a nodule (*b,d,g,h*) on *har1* (*a,b,e,g*) and MG-20 (*c,d,f,h*) hosts. Symbiont effectiveness (*a,c*) and fitness (*b,d*) were compared between derived symbionts and their corresponding ancestor using the 4wpi harvest and analysed using Welch's two sample *t*-test. Asterisks (*) indicate *p* < 0.05 and ** indicate *p* < 0.01 for *t*-tests. # represents *p* < 0.05 for ANCOVAs. Growth rate of host (*e,f*) and symbiont (*g,h*) between harvests were compared for ancestral and derived symbionts with an ANCOVA. Warm colors are used for the *har1* experiments and cool colors are used for the MG-20 experiments. (Online version in colour.)

missense mutation in an aldo/keto reductase gene, which is suggested to affect nodule development [42,43]. In both replicate populations, deletions were uncovered in a gene predicted to encode a polymerase and also within a likely *nifD* pseudogene [44] (electronic supplementary material, table S4). We are unable to dissect effects of these individual mutations since hosts were inoculated with evolved populations that included these diverse mutants. Moreover, none

of these mutations were detected at or above 50% frequency, and thus could not be confidently called in the pooled sequences of the derived populations. Pooled sequence data from the evolved CE3 populations (whether on the *har1* or MG-20 hosts) included no confident calls of fixed mutations in annotated, protein-coding genes, and instead included nucleotide polymorphisms or insertions/deletions uncovered in transposases and insertion sequence elements, many

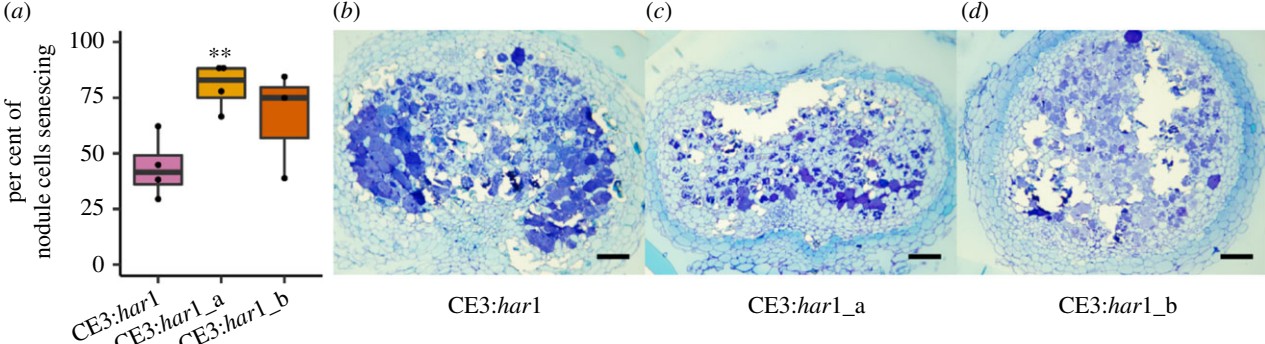

**Figure 3.** Nodule structure of ancestral and derived CE3 infected nodules. The mean number of nodule cells showing signs of senescence (*a*) was determined using visual appearance of Toluidine Blue O staining for the ancestral CE3 symbiont (*b*) and derived CE3:*har1*_a (*c*) and CE3:*har1*_b (*d*) symbionts. Scale bars are 100 μm (*e*–*g*). Asterisks indicate $p < 0.01$ using Welch's two sample *t*-test between the denoted derived symbiont and the ancestral symbiont determined. (Online version in colour.)

**Table 2.** Population parameters of rhizobia during experimental evolution.

| experiment[a] | total number of plants | total number of nodules formed | infection bottleneck size | *in planta* population size | *in planta* generations | *in vitro* generations | *in planta/ in vitro* generations |
|---|---|---|---|---|---|---|---|
| CE3:MG-20_a (15) | 253 | 559 | $39 \pm 1.4$ | $2.07 \times 10^8 \pm 1.33 \times 10^7$ | 11 394 | 128 | 89 |
| CE3:MG-20_b (15) | 245 | 660 | $47 \pm 2.4$ | $1.72 \times 10^8 \pm 7.67 \times 10^6$ | 13 821 | 137 | 101 |
| CE3:*har1*_a (15) | 255 | 1895 | $134 \pm 5.2$ | $3.19 \times 10^8 \pm 1.31 \times 10^7$ | 37 350 | 121 | 309 |
| CE3:*har1*_b (15) | 260 | 2258 | $160 \pm 6.8$ | $2.65 \times 10^8 \pm 1.18 \times 10^7$ | 44 754 | 130 | 345 |
| NGR:MG-20_a (15) | 278 | 1208 | $85 \pm 4.1$ | $4.28 \times 10^8 \pm 2.48 \times 10^7$ | 23 329 | 109 | 214 |
| NGR:MG-20_b (15) | 289 | 1464 | $103 \pm 5.1$ | $2.24 \times 10^8 \pm 1.09 \times 10^7$ | 25 586 | 126 | 203 |
| NGR:*har1*_a (10) | 448 | 4381 | $464 \pm 35.2$ | $1.92 \times 10^8 \pm 2.50 \times 10^7$ | 70 694 | 95 | 746 |
| NGR:*har1*_b (10) | 454 | 3622 | $384 \pm 29.6$ | $2.89 \times 10^8 \pm 2.71 \times 10^7$ | 69 118 | 94 | 733 |

[a]Experiments are categorized by the symbiont (CE3, NGR), host (MG-20, *har1*) and replicate (a,b). Number of passages completed is listed in parentheses.

inferred with low confidence (electronic supplementary material, table S4).

We examined variation in read depth of plasmid sequences between ancestors and derived populations, because plasmid loss was detected for the p42f plasmid in one of the derived isolates of CE3 that evolved on *har1* (table 3). Each of the plasmids appear to be single copy in the CE3 ancestor strain based on relative read depth, meaning that fluctuations downward in pooled samples are likely to reflect loss in some strains rather than variation in copy number. Evidence from the pooled samples suggests that plasmid p42f fluctuated in frequency in members of communities over the experiment, but its number appeared to recover by the final passage (electronic supplementary material, figure S2). Moreover, plasmid p42a exhibited significantly reduced copy number in all the derived populations of CE3 (on both *har1* and MG-20 hosts), which is interesting given that loss of this plasmid is associated with an increase in competitiveness for nodulation [45].

## 4. Discussion

Using experimental evolution, we simulated an agronomic lifecycle in rhizobia where they recurrently interacted with the same *L. japonicus* genotypes, similar to settings where crops are replanted in fields over multiple seasons. Given the capacity of host legumes to preferentially reward beneficial strains, we predicted that rhizobia would evolve to provide enhanced benefits to hosts [3,7,9]. Among the four symbiont-host genotype combinations that we tested, enhanced benefit evolved only in the two CE3:*har1* replicates, suggesting that both symbiont and host mechanisms impact this outcome. Each rhizobia strain had a different deficiency in association with *L. japonicus*, and thus faced distinct hurdles to evolve enhanced benefits. CE3 causes nodule senescence at 3 wpi in *L. japonicus* [26], and the evolution of greater host benefits was expected via mutations that delay instigation of nodule senescence, thus extending the period of nitrogen fixation. Although we could not resolve changes in nodule development directly, our results suggest that CE3 evolved to shift nodule senescence back by approximately a week, as the enhanced benefits of evolved CE3 populations were significant at 4 wpi—the timepoint at which we imposed selection—but were eliminated by 6 wpi (table 1 and figure 2). Moreover, senescence was observed in a higher frequency of nodule cells at 6 wpi for the evolved CE3 populations relative to their ancestor, suggesting that a delay in its onset might have caused senescence to occur in a more severe way. In opposition to CE3, NGR mutants were expected that accelerate nodule development to enhance nitrogen fixation, but instead we found negligible evidence

**Table 3.** Mutations in derived CE3 population detected from sequencing of isolates.

| gene or genome region (mutation) | replicon | frequency in CE3: *har1_a* | frequency in CE3: *har1_b* |
|---|---|---|---|
| sorbosone dehydrogenase (missense mutation) RHE_CH02735 (T > C; 2848592) | chromosome | 8/20 | 0/20 |
| alpha-L-fucosidase (missense mutation) RHE_PF00304 (C > G; 343028) | p42f | 3/20 | 0/20 |
| aldo/keto reductase (missense mutation) RHE_PE00404 (G > A; 447009) | p42e | 0/20 | 3/20 |
| nitrogenase molybdenum-iron protein alpha chain (deletion) RHE_RS30400 (218784–218792) | p42d | 4/20 | 7/20 |
| polymerase (deletion) RHE_RS22005 (36509–36510) | p42b | 7/20 | 11/20 |
| p42f (loss of plasmid) | p42f | 0/20 | 1/20 |

for NGR evolution to enhance host growth. It is possible that mutations cannot overcome the developmental delay that NGR experiences when infecting *L. japonicus* [27]. Genotype specific differences in the timing of nodule development might explain why CE3 evolved enhanced host benefits, but NGR did not.

The host genotypes we used also differed in their response to rhizobia, and might explain why CE3 evolved enhanced benefits in *har1*, but not MG-20. The *har1* experiments likely generated a greater genetic variation of symbionts for selection to act upon due to the increased number of nodules formed. Thus, although *har1* can experience costs of hypernodulation that reduce their mean benefit from rhizobia [46], the CE3: *har1* experiments generated approximately 6× the number of rhizobia replication events *in planta* compared to CE3:MG-20, and had less severe population bottlenecks during nodulation (approx. 3×), both consistent with a greater strength of selection (table 2). Previous work also demonstrated that experimental evolution of rhizobia enhanced benefit in a host genotype specific manner, in that case depending on whether the rhizobia shared evolutionary history with the host [15].

We uncovered no evidence of adaptation to the free-living experimental phases, which would be reflected in faster *in vitro* replication (table 1). The estimated number of generations during the free-living *in vitro* phases (94–137) was substantially lower than the number of generations *in planta* (approx. 11 k–70 k), suggesting that there was little opportunity for selection on *in vitro* fitness. Experimental evolution of rhizobia that includes an *in vitro* phase has the potential to instigate conflict through selection on free-living traits that counteract symbiosis [47]. Similarly, forces shaping bacteria during environmental growth are often not aligned to the interests of the host [4,20].

Genetic changes we observed in CE3 suggest some potential mechanisms for rhizobial adaptation to the experimental setting. The mutation to sorbosone dehydrogenase could, by altering polysaccharides, provide a fitness advantage *in planta* [40,48]. For instance, rhizobia have been found to scavenge resources in intercellular space after nodule cell senescence [49]. Mutations that affect signalling could have increased infection efficiency and benefit provided, while maintaining nitrogen fixation at similar levels. For instance,

alpha-L-fucosidase is associated with nod factor production [41] and is important for *trans*-cellular infection thread development in *L. japonicus* [50]. Similarly, aldo/keto reductase has been described to broadly affect nodule regulation and development [42,43]. Finally, reduced plasmid read depth in the derived versus ancestral pooled population sequences for p42a suggests that loss of this plasmid within populations might have enhanced competition for nodulation, as previously demonstrated [45]. But lack of evidence for fixed mutations weakens these results.

Enhancing nitrogen fixation involves two independent challenges: improving intracellular survival and biological nitrogen fixation within the host [51]. Evolving the capacity to nodulate legumes and persist within host cells has been experimentally demonstrated but enhancing nitrogen fixation has proven more difficult [52–54]. Although our study demonstrated the enhancement of rhizobial benefits to hosts in an experimental evolution system, we uncovered no evidence that nitrogen fixation changed significantly, suggesting that evolution reduced costs of rhizobial infection rather than enhancing nitrogen fixation. We speculate that the reduced costs might have been driven by the change in the timing of nodule senescence that we observed (figure 3). Experimental evolution of enhanced rhizobial benefit was also demonstrated in *Ensifer meliloti* rhizobia that nodulate *Medicago truncatula*, but that experiment did not measure nitrogen fixation, so it is unclear if it changed over the course of the experiment [15]. The problem of enhancing nitrogen fixation remains largely unresolved, but the current approaches suggest that both the host and symbiont genotypes must be considered in concert.

Data accessibility. Data available from the Dryad Digital Repository: https://doi.org/10.5061/dryad.ksn02v73h [55].

Authors' contributions. J.L.S.: conceptualization, formal analysis, funding acquisition, writing-original draft, writing-review and editing; K.W.Q.: conceptualization, formal analysis, writing-original draft, writing-review & editing; A.J.W. and J.H.C.: formal analysis, writing-review and editing; P.C., H.-H.L., J.T., F.S. and R.J.: methodology. All authors gave final approval for publication and agreed to be held accountable for the work performed therein.

Competing interests. We declare we have no competing interests.

Funding. This work was supported by the National Science Foundation under grant no. DEB-1738009 to J.L.S. and DEB-1738028 to J.H.C. J.L.S. was also supported by the USDA under grant no. CA-R-EEOB-5200-

H. The funders had no role in study design, data collection and analysis, decision to publish, or preparation of the manuscript.

Acknowledgements. We thank the reviewers and editor for constructive comments and suggestions. We thank Dale Noel for providing CE3, the USDA National Rhizobium Germplasm Resource Collection for NGR, LegumeBase for MG-20, and Masayoshi Kawaguchi for *har1*. We also thank the Department of Botany and Plant Pathology at OSU for supporting the computing infrastructure.

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
